# Mitochondrial DNA Instability Supersedes Parkin Mutations in Driving Mitochondrial Proteomic Alterations and Functional Deficits in Polg Mutator Mice

**DOI:** 10.3390/ijms25126441

**Published:** 2024-06-11

**Authors:** Andrew J. Trease, Steven Totusek, Eliezer Z. Lichter, Kelly L. Stauch, Howard S. Fox

**Affiliations:** 1Department of Neurological Sciences, University of Nebraska Medical Center, Omaha, NE 68198, USA; andrew.trease@unmc.edu (A.J.T.); stotusek@gmail.com (S.T.); kelly.stauch@unmc.edu (K.L.S.); 2Computational Biomedicine Section, Department of Medicine, Boston University School of Medicine, Boston, MA 02118, USA; elicht@bu.edu

**Keywords:** Parkin, Parkinson’s disease, metabolism, mitochondria, mitophagy, aging

## Abstract

Mitochondrial quality control is essential in mitochondrial function. To examine the importance of Parkin-dependent mechanisms in mitochondrial quality control, we assessed the impact of modulating Parkin on proteome flux and mitochondrial function in a context of reduced mtDNA fidelity. To accomplish this, we crossed either the Parkin knockout mouse or ParkinW402A knock-in mouse lines to the Polg mitochondrial mutator line to generate homozygous double mutants. In vivo longitudinal isotopic metabolic labeling was followed by isolation of liver mitochondria and synaptic terminals from the brain, which are rich in mitochondria. Mass spectrometry and bioenergetics analysis were assessed. We demonstrate that slower mitochondrial protein turnover is associated with loss of mtDNA fidelity in liver mitochondria but not synaptic terminals, and bioenergetic function in both tissues is impaired. Pathway analysis revealed loss of mtDNA fidelity is associated with disturbances of key metabolic pathways, consistent with its association with metabolic disorders and neurodegeneration. Furthermore, we find that loss of Parkin leads to exacerbation of Polg-driven proteomic consequences, though it may be bioenergetically protective in tissues exhibiting rapid mitochondrial turnover. Finally, we provide evidence that, surprisingly, dis-autoinhibition of Parkin (ParkinW402A) functionally resembles Parkin knockout and fails to rescue deleterious Polg-driven effects. Our study accomplishes three main outcomes: (1) it supports recent studies suggesting that Parkin dependence is low in response to an increased mtDNA mutational load, (2) it provides evidence of a potential protective role of Parkin insufficiency, and (3) it draws into question the therapeutic attractiveness of enhancing Parkin function.

## 1. Introduction

Mitochondria, of which a key and most well-known function is adenosine triphosphate (ATP) production, are also critical organelles for several other important physiological processes [1,2,3]. Among these are regulating apoptosis, maintaining REDOX balance, and regulating ion homeostasis [1,3,4,5]. Biogenesis, fission/fusion dynamics, and turnover of damaged/defective mitochondria by mitophagy all contribute to mitochondrial quality control [6,7,8,9,10,11]. Robust maintenance of mitochondrial quality is essential for the mitigation of age and disease-related mitochondrial dysfunction to maintain metabolic and bioenergetic function [12,13]. Accumulation of defective mitochondria and impaired mitochondrial quality control are common in neurodegeneration, where impaired mitochondrial function, among other mechanisms, is thought to be a key driver of synaptic failure [14,15]. In line with this, impaired mitophagy is reported in several neurodegenerative conditions, including Parkinson’s disease (PD) [16,17,18].

Though multiple mitophagy pathways exist, the most well described and relevant to neurodegeneration is the phosphatase and tensin homologue (PTEN)-induced kinase 1 (PINK1)/Parkin-dependent pathway [19,20,21,22,23]. Induction of PINK1/Parkin-dependent mitophagy is initiated by mitochondrial damage (e.g., loss of membrane potential), causing fragmentation of mitochondrial networks and outer mitochondrial membrane (OMM) accumulation of PINK1 [24,25,26,27]. Parkin—an E3 ubiquitin-ligase—is recruited and activated by PINK1 and polyubiquitinates OMM proteins that recruit autophagic machinery, promoting the formation of mitolysosomes and degrading the mitochondria [21,23,28]. Loss-of-function Parkin mutations are common genetic drivers of heritable early-onset forms of Parkinson’s disease, and thus Parkin and its role in disease are subjects of intense research [29,30,31,32,33,34].

Despite the linkage of Parkin loss-of-function mutations and impaired mitophagy to neurodegeneration in PD patients and in vitro evidence of Parkin’s role, faithful in vivo reproduction of this phenomenon has had limited success, attributed to difficult documentation of mitophagy in vivo relative to its in vitro experimental induction [35]. Although age-dependent induction of mitophagy has been shown in Drosophila, outcomes of modulating Parkin expression are inconsistent [13,36,37]. In mice, inducing mitophagy and replicating a degenerative phenotype is inconsistent. Typically, monogenic Parkin-deficient mouse models do not show degenerative phenotypes [38,39,40]. By crossing the Parkin knockout (PKO) line to the Polg mitochondrial mutator mouse, bearing a point mutation (D257A) in the proofreading domain of DNA polymerase gamma, Pickrell et al. induced a degenerative phenotype [41], although more recently, conflicting results have been reported [42]. Polg mutator mice exhibit premature aging, progressive accumulation of mitochondrial DNA (mtDNA) mutations, elevated oxidative stress, metabolic dysfunction, and mitochondrial damage [43,44,45]. Notably, when crossed with protein deglycase DJ-1-deficient mice, Polg mutator mice do not develop a degenerative phenotype. Thus, if there is an effect, it is specific to Parkin [46].

In this study, we investigated how altering Parkin expression (Parkin knockout, PKO) or activity (disinhibited mutant ParkinW402A knock-in, W402A) affects the proteomes and bioenergetic function of liver mitochondria and synaptic terminals under mtDNA stress. Generating PolgPKO and PolgW402A mice, we used mass spectrometry with longitudinal isotopic labeling to measure protein half-lives and abundance, alongside Seahorse metabolic flux assays examining bioenergetic function. Our study confirms previously observed tissue-specific protein half-lives and expands them to include non-mitochondrial synaptic proteins [47]. Moreover, we provide support for the concept that mtDNA instability increases oxidative stress and alters metabolic pathways. Surprisingly, our study suggests impaired mtDNA quality control in the Polg mutator mouse, with or without Parkin mutation, is the primary driver of proteomic changes. Finally, we reveal tissue-specific and Parkin-dependent alterations in bioenergetics due to loss of mtDNA fidelity, indicating a potential protective role of Parkin under certain conditions. Taken wholly, our study provides a foundation for future research evaluating altered Parkin activity in neurodegeneration or other diseases.

## 2. Results

### 2.1. Liver Mitochondrial but Not Synaptosomal Protein Turnover Rates Are Affected by mtDNA Fidelity

To determine whether protein turnover was impacted by impaired mtDNA quality control in PolgD257A homozygous mutant mice (Polg), we measured protein half-lives of synaptosomes and liver mitochondria analogously to our previous study [47], with slight alterations in the labeling scheme to better assess proteins with longer half-lives (Appendix A). Mice were placed on a non-labeled synthetic diet at 124 days old for a minimum of 21 days and then transitioned to an isotopically labeled ([5,5,5-2H3]-L-leucine) diet on days 145, 156, 165, 172, or 177. At 180 days old (35, 24, 15, 8, or 3 days of isotopic labeling), tryptic peptides were generated from whole brain synaptosomes or liver mitochondria and subsequently analyzed by mass spectrometry, and protein half-lives were determined using Topograph software version 1.0 [48].

In liver mitochondria, the median half-life of 396 proteins was 3.96 days (range: 1.11–35.66) in WT animals. In Polg animals, the median half-life was 4.40 days (range: 1.11–9.33) from 519 proteins (Figure 1A, Appendix A). There were 388 proteins shared between WT and Polg, with a median log2 fold difference of 0.20 (Polg/WT, range: −0.61–0.85, interquartile 0.05–0.32) (Figure 1B). Proteins annotated as mitochondrial using MitoCarta 3.0 [49] (Mus musculus) as a reference encompassed 220 and 283 proteins from WT and Polg, respectively. Median half-lives of mitochondrial proteins were 4.32 (range: 1.30–8.02) and 5.14 (range: 1.36–7.78) days for WT and Polg, respectively (Figure 1C). A total of 217 proteins were shared between strains, with a median log2 fold difference of 0.28 (range: −0.61–0.85, interquartile 0.15–0.35) Polg/WT (Figure 1D).

In synaptosomes, the median half-lives were 15.03 (range, 2.61–81.92) days from 806 proteins and 14.52 (range, 2.55–77.17) days from 1042 proteins in WT and Polg, respectively (Figure 1E, Appendix A). Between strains, the median log2 fold change (Polg/WT) of 722 proteins shared was 0.03 (range: −2.39–0.68, interquartile: −0.07–0.122) (Figure 1F). There were 227 (WT) and 292 (Polg) mitochondrially annotated proteins, with median half-lives of 20.16 (range: 4.56–49.52) and 19.81 (range: 4.59–47.35) for WT and Polg, respectively (Figure 1G). There were 217 shared mitochondrial proteins, with a median log2 fold change (Polg/WT) of 0.07 (range: −0.40–0.47, interquartile: −0.03–0.16). Taken together, this suggests that loss of mtDNA fidelity results in slower turnover of liver mitochondrial (bulk or de facto) but not bulk synaptic proteins. 

### 2.2. Loss of mtDNA Fidelity Alters Dynamics of Proteins of Metabolism and Bioenergetics

We modeled protein half-lives on a pathway scale with IPA to interrogate Polg-driven effects on protein turnover more broadly (Figure 2, Appendix A). A predominantly positive activation Z-score (i.e., longer half-life) landscape was observed for Canonical Pathways of “Metabolism” in liver mitochondria (Figure 2A). We focused on three pathways exhibiting high activation scores in liver mitochondria and synaptosomes: “Oxidative Phosphorylation”, “TCA Cycle II (Eukaryotic)”, and “Acetyl-CoA Biosynthesis I (Pyruvate Dehydrogenase Complex)”. All exhibited positive activation Z-scores in both tissues (i.e., slower turnover in Polg). In the “TCA Cycle II (Eukaryotic)” pathway, 92.3% (liver) and 93.3% (synaptosomes) of detected proteins were turned over more slowly in Polg than in WT, with an average increase in half-life of 0.98 ± 0.56 and 1.42 ± 1.18 days, respectively (Figure 2B). Half-lives of all detected “Acetyl-CoA Biosynthesis (Pyruvate Dehydrogenase Complex)” proteins were increased in Polg animals compared to WT, independent of tissue source (liver: 1.10 ± 0.52 days, synaptosomes: 3.65 ± 3.08 days; Figure 2C). Although many “Oxidative Phosphorylation” proteins had longer half-lives in both tissues, the pathway only displayed a strong positive activation Z-score in liver mitochondria (Figure 2D). This is mostly likely due to faster turnover of Complex IV subunits in Polg synaptosomes (Figure 2D). Half-lives of subunits of Complex IV encoded by mtDNA were unchanged in both liver mitochondria (Mtco1) and synaptosomes (Mtco1 and Mtco2; Figure 2D; Complex IV, red). This was contrasted by the Complex I and III subunits (Mtnd1, Mtnd4, and Mtcyb) in synaptosomes and the Complex V subunit Mtatp8 in both liver mitochondria and synaptosomes, which were all characterized by Polg-dependent increased half-lives (Figure 2D; Complex I, III, V; red).

### 2.3. Impaired mtDNA Fidelity Influences Oxidative Phosphorylation and Electron Transport Protein Expression

To determine whether differences in protein turnover rates were accompanied by altered abundance, we applied our previously validated label-free quantitative approach to isolated liver mitochondria and synaptosomes [47]. We initially identified 2338 proteins in liver mitochondria and 3784 in synaptosomes, of which 1120 and 1600, respectively, were reproducibly quantifiable. Comparing Polg to WT, we found 65 differentially expressed (DE) proteins in the liver and 104 in synaptosomes, with 34 upregulated and 31 downregulated in the liver and 78 upregulated and 26 downregulated in synaptosomes (Figure 3A,B, Appendix A). Applying a threshold requiring valid values in at least 50% of samples per genotype, we identified 18 upregulated and 16 downregulated proteins in the liver and 30 upregulated and 8 downregulated proteins in synaptosomes (Figure 3A,B, dark dots). Mitochondrially annotated DE proteins passing the missing value threshold [49] are highlighted (Figure 3A,B, black outline/text labels). Proteins detected in both tissues and determined as DE (missing value threshold aside) in either tissue (Figure 3C, black—liver, blue—synaptosomes, or red—both) were hierarchically clustered. Notably, heme-binding protein 1 (Hebp1) expression was elevated in both liver mitochondria and synaptosomes. Pathway analysis for “Canonical Pathways” of “Metabolism” predicted by IPA revealed strong negative activation Z-scores (Figure 3D, Appendix A) in both tissues for “Oxidative Phosphorylation” in both the liver (−4.73) and synaptosomes (−3.27), indicating reduced expression in Polg. Several catabolic pathways displayed similar degrees of positive activation Z-scores in liver mitochondria and synaptosomes, including those associated with degradation of catecholamines and indoleamines.

### 2.4. Influence of Parkin Expression or Activity on Proteome Dynamics in Polg Animals

#### 2.4.1. Parkin Deficiency Reduces Turnover of Liver Mitochondrial Proteins

Previously, we reported that loss of Parkin led to tissue-specific alterations in the expression of select mitochondrial proteins under homeostatic conditions [47]. Due to Parkin’s role in mitochondrial quality control under stressed conditions (mitophagy), we sought to determine the effect of Parkin insufficiency in the context of mtDNA instability. Thus, we bred the Parkin knockout (PKO) and Polg mutator lines to generate PolgD257A/PKO (PolgPKO) double mutants (Appendix A). Consistent with genotyping, PolgPKO mice lack full-length Parkin protein in synaptosomes (Appendix A). Following the above approach, we compared half-lives of PolgPKO and Polg mitochondria and synaptosomes. Half-lives were determinable in 535 (median: 4.56 days; range: 1.09–35.27) proteins in liver mitochondria from PolgPKO animals (compared to 4.40 days in Polg) (Figure 4A, Appendix A). Directly comparing the 491 proteins shared between Polg and PolgPKO, we determined the median log2 fold change was 0.03 (range: −0.48–0.52, interquartile −0.06–0.12) (Figure 4B). There were 290 mitochondrially annotated proteins with quantified half-lives (median: 5.24 days; range 1.28–7.63) (Figure 4C). Median half-life and ranges were similar between Polg and PolgPKO in bulk and mitochondrially annotated proteins (Figure 4B,C). Direct comparison of 273 proteins between Polg and PolgPKO resulted in a median log2 fold change (PolgPKO/Polg) of 0.04 (range: −0.48–0.52, interquartile: −0.02–0.11) (Figure 4D). Z-scores were calculated against the mean of half-lives between genotypes for each protein (Figure 4E).

In synaptosomes of PolgPKO animals, we quantified the half-lives of 1067 proteins. Comparable to Polg synaptosomes (14.52 days), the median half-life was 14.44 days (range: 2.72–284.4) (Figure 4F, Appendix A). The median log2 fold change half-life of 978 proteins in both Polg and PolgPKO was 9.3 × 10^−5^ (range: −0.90–2.13, interquartile: −0.08–0.08) (Figure 4G). Annotation of synaptic proteins of PolgPKO revealed 298 mitochondrial proteins, with a median half-life of 19.77 (range: 4.56–44.57) (Figure 4H). A total of 280 mitochondrial proteins overlapped between Polg and PolgPKO, with a median log2 fold change of −0.03 (range: −0.90–0.40, interquartile: −0.09–0.07) (Figure 4I). Interestingly, PKO induced a right shift in the half-lives of liver but not synaptosomal proteins. Z-scores were calculated against the mean of half-lives between genotypes for each protein (Figure 4J).

The Z-score analysis confirmed that the majority of changes are driven by Polg (i.e., right shift, decreased turnover, longer half-lives), and that loss of Parkin seems to have a somewhat restorative effect on protein turnover rate in Polg mice (i.e., left shift). Interestingly, the W402A leads to an even larger shift to the left (i.e., increased turnover, shorter half-lives).

#### 2.4.2. ParkinW402A Enhances Turnover of Synaptic Mitochondrial Proteins

Parkin activity is autoinhibited by molecular interactions between two domains (REP and RING) in the closed conformation dependent on residue Trp403 (Trp402 in murine Parkin), and in vitro mutation of Trp403 to alanine leads to dis-autoinhibition, increased mitochondrial translocation, and substrate ubiquitination [50,51,52]. We created PolgD257A/ParkinW402A (PolgW402A) double mutants by crossing the Polg and ParkinW402A (W402A) lines (Appendix A). Parkin expression was slightly lower in PolgW402A synaptosomes than in WT or Polg (Appendix A). To test whether ParkinW402A promoted enhanced turnover in vivo, we then measured half-lives for 524 proteins (median: 4.48 days; range 1.04–36.25; compared to 4.40 in Polg, see above) in liver mitochondria from PolgW402A mice (Figure 4A, blue). There were 481 proteins shared between Polg and PolgW402A, with a median log2 fold change of 0.09 (range: −0.67–0.44; interquartile: −0.04–0.07) (Figure 4B, Appendix A). In PolgW402A, mitochondrially annotated proteins comprised 291 of those with half-lives, with a median half-life of 5.13 (1.21–7.94) days (Figure 4C). Polg and PolgW402A 272 proteins were comparable, with a median log2 fold change of 0.01 (range: −0.33–0.41; interquartile: −0.03–0.07) (Figure 4D).

From PolgW402A synaptosomes, the median half-life of 1048 proteins was 14.32 days (range: 2.87–119.820) (Figure 4E, Appendix A). Those comparable between Polg and PolgW402A (931 proteins) resulted in a median log2 fold change ratio (PolgW402A/Polg) of −0.04 (range: −1.11–1.64, interquartile: −0.11–0.04) (Figure 4F). Like Polg and PolgPKO synaptosomes, analysis limited to mitochondrially annotated proteins (287) revealed the half-lives increased compared to bulk synaptic proteins, with a median of 19.50 days (range: 4.13–44.57) (Figure 4G). Fewer mitochondrial proteins overlapped between Polg and PolgW402A (265) than between Polg and PolgPKO (280). There was a median log2 fold change (PolgW402A/Polg) of −0.06 (range: −1.11–0.41, interquartile: −0.12–0.02) (Figure 4H). Although neither liver mitochondrial nor bulk synaptic protein turnover rates were impacted by Parkin dis-autoinhibition, synaptic mitochondrial annotated proteins exhibited a notable left shift in PolgW402A compared to Polg (Figure 4H, blue), suggesting dis-autoinhibited ParkinW402A may increase turnover of synaptic mitochondrial proteins and partially restore a WT-like state. Mitochondrial annotated proteins exhibited longer half-lives compared to the bulk proteins, regardless of tissue.

### 2.5. Polg Is the Primary Driver of Altered Proteostasis and Tissue, Specifically Modified by Parkin

We next focused on how loss or dis-autoinhibition of Parkin impacts the global protein turnover landscape. To accomplish this, area overlap plots (Appendix A) and global Spearman correlation matrices were generated (Appendix A) for every pairwise combination of liver mitochondria or synaptosomes, which suggested most protein turnover alterations in Polg animals were Polg-driven. Correlation was high between the Polg/WT and either the double mutant or WT (PolgPKO/WT, r = 0.77; PolgW402A/WT, r = 0.84) in liver mitochondria and moderate in synaptosomes (PolgPKO/WT, r = 0.66; PolgW402A/WT, r = 0.61). This further suggested some novel changes driven by modulation of Parkin in the context of mtDNA stress when comparing the double mutant to Polg comparisons to Polg/WT (liver: PolgPKO/Polg–Polg/WT r = −0.15, PolgW402A/Polg–Polg/WT r = −0.14; synaptosomes: PolgPKO/Polg–Polg/WT r = −0.33, PolgW402A/Polg–Polg/WT r = −0.55). To identify differences that would affect metabolism, we used IPA (Figure 5A, Appendix A). Most “Metabolism” pathways revealed a trend of tissue specificity that was similar between PolgPKO and PolgW402A, with most pathways showing negative activation Z-scores (i.e., faster turnover) in synaptosomes and positive scores (i.e., slower turnover) in liver mitochondria (Figure 5B). We targeted the individual protein half-lives of the same three metabolic pathways exhibiting robust activation Z-scores in the Polg/WT comparison (see Figure 2). Contrasting the Polg/WT comparison, in synaptosomes, “TCA Cycle II (Eukaryotic)” and “Acetyl-CoA Biosynthesis (Pyruvate Dehydrogenase Complex)” exhibited negative activation Z-scores in either PolgPKO/Polg or PolgW402A/Polg, whereas this was positive in the liver (Figure 5B).

The change in half-life (ΔHalf-life) of individual “TCA Cycle II (Eukaryotic)” proteins from Polg/WT (Figure 5C, black) to that of either PolgPKO/Polg (Figure 5C, red) or PolgW402A/Polg (Figure 5C, blue) exhibited similar patterns for each double mutant in either tissue. Exceptions to this were isocitrate dehydrogenase [NAD]+ subunit gamma (IDH3G), dihydrolipoyllysine-residue succinyltransferase (DLST), citrate synthase (CS), and succinate dehydrogenase subunit A (SDHA), of which there was partial rescue in the liver mitochondria of PolgW402A animals (Figure 5C). More interesting, however, is the difference observed between the liver and synaptosomes. In the liver, only 23% and 46% of detected TCA cycle proteins from the PolgPKO/Polg and PolgW402A/Polg comparisons exhibited reversed directionality compared to Polg/WT, whereas in synaptosomes, there was 80% reversal in both comparisons (Figure 5C).

A similar phenomenon was observed for the proteins in the “Acetyl-CoA Biosynthesis (Pyruvate Dehydrogenase Complex)” pathway (Figure 5D). Interestingly, this pattern was contrasted in the “Oxidative Phosphorylation” pathway, where IPA predicted positive activation Z-scores for both PolgPKO/Polg and PolgW402A/Polg in the liver as well as PolgPKO/Polg but not PolgW402A/Polg in synaptosomes (Figure 5B). The Δ half-life values of the individual proteins were reversed in the “Oxidative Phosphorylation” pathway for 23.6%, 35.1% (liver: PolgPKO/Polg, PolgW402A/Polg), and 70.8% (synaptosomes: PolgPKO/Polg) of proteins compared to the corresponding Polg/WT value, whereas only 8.9% were reversed for PolgW402A/Polg in synaptosomes (Figure 5D. Individual complexes exhibited the same trend, with the synaptic PolgPKO/Polg comparison consistently showing the highest percentage of subunits with reversed directionality (Complex I—68.2%, Complex II—33.3%, Complex III—57.2%, Complex IV—85.7%, and Complex V—88.9%), and the synaptic PolgW402A/Polg comparison showing the least (Complex I—2.2%, Complex II—0.0%, Complex III—14.3%, Complex IV—0.0%, and Complex V—22.2%). 

### 2.6. Parkin’s Influence on Protein Expression Is Greater in Liver Mitochondria Than Synaptosomes

Next, we measured protein abundance in the double mutants compared to Polg alone. From 1120 quantified proteins, 88 were DE in PolgPKO animals compared to Polg; 36 were downregulated and 52 upregulated, of which 21 and 33 passed our missing value cutoff, respectively (Figure 6A, Appendix A). Of these, 15 of the downregulated proteins and 4 of the upregulated ones were mitochondrially annotated proteins (Figure 6A, black outline). Expression of two mitochondrial proteins (Figure 6A, red text) was reversed in PolgPKO/Polg compared to Polg/WT, and only one (Figure 6A, blue text) exhibited a synergistic effect. In PolgW402A liver mitochondria, there were 72 DE proteins compared to Polg, including 59 downregulated and 13 upregulated proteins (38 and 6 passing missing value cutoff, respectively; Figure 6B). Six of the downregulated and two of the upregulated proteins were annotated as mitochondrial (Figure 6B, black outline). Six proteins were DE (passing missing value cutoff) in both PolgPKO/Polg and PolgW402A/Polg comparisons.

Of 1600 quantified synaptic proteins, 69 were DE in PolgPKO compared to Polg animals, with 34 downregulated and 35 upregulated proteins; 8 and 13 passed the missing value threshold, respectively (Figure 6C, dark dots). Three downregulated and five upregulated proteins were annotated mitochondrial (Figure 6C, black outline). Comparing PolgW402A to Polg, there were 67 DE proteins, including 41 downregulated and 26 upregulated ones and 10 with sufficient valid values in either direction (Figure 6D; dark dots). One downregulated and four upregulated proteins were annotated mitochondrial (Figure 6D, black outline). Expression of two synaptic DE proteins in PolgPKO/Polg (both upregulated; Figure 6C, red text) and three in PolgW402A/Polg (one down and two upregulated; Figure 6D, blue and red text) was reversed compared to the Polg/WT. Only one protein was DE in both PolgPKO/Polg and PolgW402A/Polg and was upregulated in both cases.

### 2.7. Pathway Analysis Reveals Parkin Fine Tunes Polg-Driven Protein Abundance Effects

IPA of Canonical Pathways of “Metabolism” (Figure 7A, Appendix A) revealed a tissue-specific effect of Parkin modulation. In liver mitochondria, PolgPKO and PolgW402A exhibited opposing pathway activation compared to Polg animals. Phosphoinositide metabolism had similar patterns in liver mitochondria of PolgPKO and PolgW402A compared to Polg; however, Parkin insufficiency led to strong activation in synaptosomes of PolgPKO animals compared to Polg. In synaptosomes, most pathways of “Metabolism” exhibited positive activation Z-scores that were generally comparable between PolgPKO and PolgW402A. “tRNA Charging”, “Triacylglycerol Degradation”, and “TCA Cycle II (Eukaryotic)”; however, they were differentially affected in PolgPKO and PolgW402A synaptosomes relative to Polg.

Focusing on the same metabolic pathways, we assessed whether altered half-lives occurred in unison with changes in protein abundance by analyzing log2 fold change expression ratios of proteins quantified from “TCA Cycle II (Eukaryotic)” (Figure 7B), “Acetyl-CoA Biosynthesis (Pyruvate Dehydrogenase Complex)” (Figure 7C), and “Oxidative Phosphorylation” (Figure 7D, subdivided by respiratory complex). Maintaining our missing value cutoff, we focused on proteins exhibiting a ≥|0.5| log2 fold change in the respective double mutant compared to Polg to identify proteins regulated by Parkin. In the “TCA Cycle II (Eukaryotic)” pathway, 15 of 16 proteins were quantified in either liver mitochondria or synaptosomes. Only one in liver, uniquely impacted by PolgW402A, was DE (Figure 7B). Two synaptic “TCA Cycle II (Eukaryotic)” proteins were DE (Figure 7B); however, they exhibited differing Parkin-dependent effects. Five of six “Acetyl-CoA Biosynthesis (Pyruvate Dehydrogenase Complex)” pathway proteins were quantified in liver mitochondria and synaptosomes (Figure 7C). In liver mitochondria, two were affected by Parkin status (Figure 7C). Among the respiratory complexes of “Oxidative Phosphorylation”, Complex I was the most affected in either tissue, with four and nine DE proteins in liver mitochondria or synaptosomes, respectively (Figure 7D). No Complex II proteins met analysis requirements. Only one Complex III protein was DE based on Parkin status in liver mitochondria, and two were in synaptosomes. For Complex IV, Parkin impacted one protein in liver mitochondria and three in synaptosomes.

### 2.8. Polg Mutator Mice Exhibit Tissue-Specific Bioenergetic Capacity Impacted by Parkin Status

To evaluate the functional consequences of altered turnover and abundance, isolated synaptosomes or liver mitochondria from each strain were evaluated on a Seahorse Metabolic Flux Analyzer by Mito Stress Test (synaptosomes; Figure 8A–C) or Electron Flow Assay (liver mitochondria; Figure 8D,E). In synaptosomes, there was no significant difference in respiration between WT and Polg animals. A reduced oxygen consumption rate (OCR) was observed in Polg animals lacking Parkin (Figure 8B), correlating with significantly impaired maximal and spare respiratory capacities (Figure 8C). Maximal respiratory capacity was also impaired in synaptosomes of PolgW402A mice compared to WT animals (Figure 8C).

Electron transport chain (ETC) efficiency was tested in isolated liver mitochondria of each strain using the Electron Flow Assay. Contrasting synaptosomes, liver mitochondria from Polg animals exhibited significantly impaired ETC function at Complexes II and IV, although increased variability in succinate-driven respiration (Complex II) suggests this is likely due to Complex IV impairment rather than Complex II impairment (Figure 8D,E). Surprisingly, PolgPKO mice showed no ETC dysfunction, indicating a potential rescue from mtDNA fidelity loss (Figure 8E). PolgW402A liver mitochondria did not significantly differ from WT in respiration. Complex II function was partially rescued in PolgW402A compared to Polg, albeit to a lesser degree than in PolgPKO. In left ventricular mitochondria, PolgPKO animals had similarly improved electron flow efficiency at Complexes II and IV (Appendix A). Contrasting liver mitochondrial function, dis-autoinhibited Parkin did not have a detrimental effect on electron flow in ventricles.

## 3. Discussion

Our study reveals loss of mtDNA fidelity drives altered protein abundance, turnover rate, and metabolic function in liver mitochondria and synaptic terminals in vivo. Consistent with our previous report [47], we find divergent protein turnover rates between liver mitochondria and synaptosomes. Additionally, our results demonstrate altering Parkin expression (PKO) or activity (W402A) elicits tissue-specific effects, suggesting a surprising protective role for Parkin deficiency in tissues with rapid mitochondrial turnover. Specifically, we show mitochondria from the liver are more susceptible to altered proteostasis from impaired mtDNA fidelity than synaptic mitochondria, exhibiting a greater relative increase in median half-life. Conversely, though alterations in synaptic protein half-lives are more subtle, the synaptic proteome is more vulnerable to Polg-driven effects on protein expression. These turnover and abundance changes affected metabolic pathways associated with energy production. Critically, our findings limit the appeal of modulating Parkin activity as an intervention for mtDNA-associated disorders. 

Fidelity in mitochondrial genome replication is essential for mitochondrial homeostasis, ensuring proper mitochondrial function to fulfill metabolic needs, limit oxidative damage, and regulate cell survival [53]. Dysfunctional DNA polymerase gamma is a key driver of mitochondrial dysfunction, implicated in several human metabolic and neurological conditions, including retinal degeneration as well as forms of epilepsy, Parkinsonism, and early-onset familial PD [54,55,56]. Though not directly linked to sporadic PD, the mitochondrial consequences of mtDNA instability are strikingly similar to those in PD, including Complex I and IV dysfunction [57,58]. Additionally, PD is associated with acetyl-CoA synthesis and TCA cycle dysfunction. When combined with respiratory chain dysfunction, this can exacerbate bioenergetic deficits [59,60,61,62,63]. 

Our results demonstrate loss of mtDNA fidelity resulted in disruptions in turnover and abundance of proteins involved in acetyl-CoA synthesis, the TCA cycle, and oxidative phosphorylation in male Polg mice aged to 6 months. In both liver mitochondria and synaptosomes, these changes present as prolonged half-lives across all three pathways, accompanied by increased abundance of TCA cycle and acetyl-CoA synthesis proteins. Conversely, oxidative phosphorylation protein abundance was generally lower in Polg animals despite increased half-lives. Reduced oxidative phosphorylation is known to occur in Polg mice starting in proximity to the age we investigated [45], the long-term consequences of which are not well investigated in this particular model. Though not directly assessed, concomitant induced expression of TCA cycle and acetyl-CoA pathway proteins and inhibition of oxidative phosphorylation suggests increased glycolytic flux and is consistent with reports associating mtDNA depletion or increased mutational load with increased glycolysis [63,64,65]. In support of this, we observed increased expression of “Glycolysis I” proteins in both liver and synaptosomes of Polg animals (see Figure 3D). Furthermore, the impaired electron flow efficiency in liver mitochondria is congruent with a compensatory shift towards glycolysis from elevated ROS production, a promoter of glycolytic transcriptional activation [66]. Indeed, in PolgPKO compared to Polg mice, we uncovered proteomic changes suggestive of increased glycolysis with decreased turnover of proteins of “Glycolysis I” (see Figure 5A). A probable explanation for this shift in metabolism is a compensatory mechanism for the loss of impairment of oxidative phosphorylation. In PD, upregulation of glycolysis is thought to be a mechanism allowing dopaminergic neurons to cope with bioenergetic deficits [67]. One notable difference between liver mitochondria and synaptosomes was the Polg-driven effects on Complex IV subunit half-lives. Though increased in Polg livers, they were decreased in synaptosomes, despite which both tissues show a similar net effect on abundance, the significance of which is difficult to interpret.

Importantly, though there were tissue- and Parkin-dependent effects on both turnover and abundance of select proteins, they were complex and contrary to anticipation. We hypothesized that Parkin deficiency would exacerbate Polg-driven proteomic changes, increasing the median protein half-life and overall protein abundance. Conversely, we expected ParkinW402A to in part mitigate the Polg-driven effects. The effects of modulating Parkin expression or activity on the three metabolic pathways we examined in detail were, however, complex. In liver mitochondria, Parkin deficiency and dis-autoinhibition enhanced Polg-driven effects on protein turnover, whereas in synaptosomes, both exhibited a trend to WT restoration, though moderate. The Z-score analysis confirmed that the majority of protein turnover changes are driven by Polg, and that loss of Parkin seems to have a somewhat restorative effect, which may be due to the induction of alternative quality control pathways. Further, the W402A reveals proteins that are likely selectively turned over by Parkin, as dis-autoinhibition of Parkin leads to their increased turnover. This could be due to direct turnover via Parkin or via indirect mechanisms (i.e., Parkin regulates pathways important for their synthesis or degradation).

The most interesting finding of our study was the unexpected bioenergetic functional rescue seen with Parkin insufficiency in liver and heart mitochondria and the detrimental effect of dis-autoinhibition in the liver mitochondria in the Polg context. We observed reduced electron flow efficiency in liver mitochondria of Polg animals, which was restored to WT levels in PolgPKO animals. Though ventricular mitochondria were not significantly impaired by loss of mtDNA fidelity, loss of Parkin improved electron flow efficiency beyond WT levels. Traditionally, loss of Parkin is thought to promote increased mitochondrial dysfunction [39,41,68]; however, numerous recent reports, including our study, contest this [36,42,69,70,71]. Previously, our lab reported rats lacking Parkin do not exhibit functional deficits in striatal synaptic mitochondria [72]. Additionally, in a recent preprint, Filograna et al. report that genetic ablation of Parkin did not impair activity of oxidative phosphorylation enzymes in mitochondria isolated from brain, heart, or skeletal muscle [71]. Interestingly, though not reaching significance, Complex I and Complex I/Complex III activity was modestly elevated in heart mitochondria of Parkin-null mice compared to WT. Moreover, ATP production of Parkin-null PD patient-derived skeletal muscle mitochondria was modestly increased when stimulated with substrates specific to Complex I + II (glucose and succinate) or Complex IV (ascorbate/TMPD) [71]. Scott et al. similarly reported no significant differences in the activity of Complex I or Complex IV of Parkin-null mice [42]. Notably, both studies also investigated the effects of Parkin insufficiency in the Polg mutator mouse, concluding that loss of Parkin did not exacerbate the Polg phenotype [42,71]. Despite this evidence, the possibility does exist that the bioenergetic rescue effect we observed may be the result of Parkin-independent effects.

The effect of dis-autoinhibiting Parkin (ParkinW402A) in the case of Polg mutation is more cryptic. In tissues of rapid turnover, the effect of overactive Parkin was detrimental (liver mitochondria) or non-existent (heart mitochondria). Similarly, in synaptic terminals, there was little to no effect of overactive Parkin in the context of Polg mutation. To date, only two other studies have examined ParkinW402A in vivo [73,74], neither investigating bioenergetic function. One focused on the degree of mitophagy induction of the ParkinW402A mutant after insult with lipopolysaccharide in the heart, reporting no improvement in the induction of mitophagy nor activation of Parkin compared to WT [74]. The second, conducted by our lab, examined the effects of both loss and dis-autoinhibition of Parkin on mtDNA integrity in the context of the Polg model [73]. We found that loss of Parkin led to mtDNA mutations and increased control region multimers (CRMs); however, dis-autoinhibition of Parkin was associated with a higher degree of heteroplasmy and a greater prevalence of deleterious mutations [73]. Our results suggest two things: (1) that loss of Parkin improves mitochondrial function in the context of Polg mutation, at least in the liver and heart; and (2) there is limited therapeutic attractiveness to using enhanced Parkin activity to combat neurodegeneration. One possible explanation for our results may lie in differences in the number of CRMs, which could compensate for the development of Polg-mediated detriments in the PolgPKO but not PolgW402A mice [73]. If this is the case, a threshold likely exists at which point CRMs would, as we previously hypothesized [73], no longer compensate; however, this would require investigation at additional ages.

In our previous report [47], we demonstrated tissue-specific turnover rates of mitochondrial proteins, with liver mitochondria showing faster turnover (median: 3.82 days) than striatal synaptic mitochondria, with considerably longer half-life (median: 27.0 days). Here, we confirm these findings, showing the median half-life of liver mitochondrial proteins in WT animals is 3.96 days and also validating the labeling paradigm modifications. Interestingly, the median synaptosomal half-life was 15.03 days, shifting to 20.16 days when focusing on mitochondrial proteins. This contrasted with our previous findings in striatal synaptic mitochondria, where the median half-life was 25.79 (bulk) or 27.04 days (mitochondrially annotated). Several explanations could explain this discrepancy. First, our previous shorter metabolic labeling period may have underestimated amino acid recycling, introducing greater variability at longer half-lives [48]. Secondly, there might be age-dependent differences (3 versus 6 months), however unlikely, as declining protein turnover correlates with aging [75,76,77,78]. Consistent with this, the half-lives of liver mitochondria in this study were modestly longer than in our previous report [47]. The most likely explanation for the shorter half-lives in synaptosomes is a difference in turnover rate based on source. We previously measured half-lives of striatal synaptic mitochondria, whereas here, we measured those of whole brain synaptosomes, likely shifting the half-life from the inclusion of additional neuronal subpopulations. A recent study supporting this reported differing protein half-lives between neurons and glia, as well as between neurons of the cortex and those of the cerebellum [79]. Importantly, taken together, these studies suggest protein turnover is, in part, spatially regulated. Furthermore, when considering the Parkin-dependent effects we observed on mitochondrial function and the association between mitochondrial dysfunction and neurodegeneration, it is possible that this spatial regulation of protein turnover contributes to selective vulnerability [80] of disease-specific neuronal subpopulations, warranting further investigation.

## 4. Materials and Methods

### 4.1. Study Design

In this study, wild-type (WT, Stock No: 000664, C57BL/6J), Parkin knockout (PKO, Stock No: 006582, B6.129S4-Prkntm1Shn/J) [39], Parkin W402A knock-in (W402A, Stock No: 029317, C57BL/6N-Prknem1Mjff/J, Taconic Biosciences, Germantown, NY, USA), and mitochondrial polymerase γ mutator (Polg, Stock No: B6.129S7(Cg)-Polgtm1Prol/J) [44] mice were obtained from The Jackson Laboratory. Heterozygous mutant Polg males were crossed with females homozygous for PKO or W402A mutations to obtain double-mutant PolgPKO or PolgW402A strains. Colonies and germline mitochondrial mutational loads were controlled by following the breeding paradigm presented in Appendix A. Mice were housed in a temperature-controlled environment, with a 12/12 light/dark cycle, and given ad libitum access to food and water throughout the study. After confirmation of genotype, three male mice of each strain (WT, Polg, PolgPKO, PolgW402A) were randomly assigned to groups by the number of days on heavy labeled feed, totaling 18 mice per strain. Liver mitochondria (*n* = 3) and synaptic nerve terminals (synaptosomes, *n* = 3) were isolated (see below) from each time point. Independent mice were used for the bioenergetic experiments totaling 13 (WT), 7 (Polg), 8 (PolgPKO), and 5 (PolgW402A). In total, 105 male mice were used in the experiments herein. For turnover calculations only, each experimental unit is a pool of all 3 mice for each condition (see Section 2.2); in all other experiments, the experimental unit is defined as a single male mouse. Experimental sample sizes were determined based on previous studies from our laboratory to (1) calculate protein turnover rates [47] and (2) appropriately represent bioenergetic differences between strains [81]. The University of Nebraska Medical Center Institutional Animal Care and Use Committee (IACUC; protocol numbers 21-089 [experimental] and 08-037 [breeding]) approved all breeding and experimental procedures described here, which were carried out following approved protocols and regulations. Study methods were conducted and are reported per the ARRIVE guidelines. Animal exclusion criteria were determined as criteria for euthanasia by experimental IACUC protocol number 21-089, the presence of gross anatomical deformities (including the presence of tumors, blocked ureters, teratomas, or organs with abnormal appearance (e.g., color of liver)) and general malady.

### 4.2. Metabolic Labeling with Heavy Leucine

At 124 days of age, mice were transitioned from standard rodent chow to a synthetic diet containing only light leucine (Teklad TD.01084, Envigo, Indianapolis, IN, USA) to standardize the tracer precursor pool to enable protein turnover measurements. After 21 days of preconditioning, mice were randomly assigned to 5 groups: 3, 8, 15, 24, or 35 days of metabolic labeling. At this point, mice were fed a leucine-deficient synthetic diet (Teklad TD.110642, Envigo) supplemented with 11.1 g/kg of deuterated [5,5,5-2H3]-L-leucine (Cambridge Isotope Laboratory, Tewksbury, MA, USA) at 177, 172, 165, 156, or 145 days of age (Appendix A). Mice were sacrificed by cervical dislocation at 180 days old, corresponding with 5 lengths of metabolic labeling: 3, 8, 15, 24, and 35 days on [2H3]-leucine feed. Mice that did not receive [2H3]-leucine feed were included to obtain body and organ weights. Body weights of mice were recorded at 30-day intervals from 60 to 180 days of age. PolgPKO mice consistently weighed less than WT counterparts, and Polg mice weighed less at 150 and 180 days of age. Though modestly smaller than WT at 60 days of age, PolgW402A were not significantly different at any other age (Appendix A). All Polg strains displayed a significant reduction in progressive body weight change from 120 to 150 days of age (4–5 months) that was most pronounced in the PolgPKO mice (Appendix A). On average, mice gained 28.3%, 20.4%, 19.4%, and 26% of their initial body weight from 60 to 180 days for WT (6.4 ± 2.4 g), Polg (4.2 ± 2.4 g), PolgPKO (3.8 ± 2.0 g), and PolgW402A (5.6 ± 1.9 g), respectively (Appendix A).

### 4.3. Subcellular Fractionation

#### 4.3.1. Liver Mitochondria

Mitochondria were isolated from livers as previously described [82]. Liver tissue was minced and subsequently homogenized using a Dounce homogenizer with 10 strokes of pestle A followed by 10 strokes of pestle B in ice-cold mitochondrial isolation buffer (MSHE + BSA) containing 70 mM sucrose, 210 mM mannitol, 5 mM HEPES, 1 mM EGTA, and 5 mg/mL fatty acid free BSA (pH 7.2), supplemented with protease inhibitor. Debris was removed from the homogenate by centrifugation at 600× *g* for 10 min, the resulting supernatant was collected, and a crude mitochondrial pellet was obtained by centrifugation at 15,000× *g* for 10 min. The crude mitochondrial pellet was resuspended in MSHE + BSA and fractionated via ultracentrifugation for 30 min at 34,750× *g* on a discontinuous density gradient of MSHE + BSA containing 26% or 60% Percoll. The band containing mitochondria at the interface of 26% and 60% was collected, diluted with MSHE + BSA, and centrifuged for 10 min at 16,700× *g*. The resulting mitochondrial pellet was washed by resuspending in mitochondrial assay solution (1x MAS) containing 70 mM sucrose, 220 mM mannitol, 10 mM KH_2_PO_4_, 5 mM MgCl_2_, 2 mM HEPES, and 1 mM EGTA and pelleted by centrifugation at 13,000× *g* for 10 min. For mass spectrometry samples, the final pellet was lysed directly by sonication in 100 mM Tris-HCl with 4% (*w*/*v*) SDS and 0.1 M DTT adjusted to pH 7.6 and heated to 95 °C for 5 min. For Seahorse experiments, the final pellet was resuspended in 1x MAS. Protein concentration was determined by Pierce 660 nm Protein Assay. Lysates for mass spectrometry were stored at −80 °C until further processing. Seahorse samples were used fresh.

#### 4.3.2. Heart Ventricular Mitochondria

The left ventricular wall and septum were dissected from hearts, minced, and homogenized using, in sequence, 5 strokes of 5 mL motor-driven Teflon Potter-Elvehjem homogenizer at ~3500 RPM, 10x strokes of pestle A, and 10x stokes of pestle B in a 2 mL Dounce homogenizer in ice-cold MSHE + BSA with protease inhibitor. Debris was removed from the homogenate by centrifugation at 1300× *g* for 3 min. From this point, heart ventricular mitochondria were isolated as described above for liver mitochondria, and the final pellet was resuspended in 1x MAS for Seahorse experiments for protein concentration determination by Pierce 660 nm Protein Assay. 

#### 4.3.3. Synaptic Terminals (Synaptosomes)

Synaptosomes were isolated as previously described with minor modifications [83]. Briefly, whole brains were dissected out, and the olfactory bulbs and cerebellum were removed. The resulting tissue was homogenized in ice-cold homogenization buffer (HB; 320 mM sucrose, 1 mM EDTA, 5 mM Tris (pH 7.4)) supplemented with protease inhibitor in a glass Dounce homogenizer fitted by 10 strokes with pestle B. Bulk debris was removed from the homogenate by centrifugation at 1000× *g* for 10 min, the supernatant was collected, the pellet was resuspended in HB, and the centrifugation was repeated. Supernatants were pooled and brought to a total volume of 6 mL with 3% Percoll containing 320 mM sucrose, 1 mM EDTA, and 5 mM Tris (pH 7.4). This suspension was then overlaid and fractionated by ultracentrifugation on a discontinuous Percoll gradient of 10% and 23% Percoll containing 320 mM sucrose, 1 mM EDTA, and 5 mM Tris (pH 7.4). After ultracentrifugation at 58,755× *g* for 12 min, the band containing synaptosomes, at the interface of 10% and 23%, was collected and diluted with ionic media (20 mm HEPES, 10 m glucose, 1.2 mM Na_2_HPO_4_, 1 mM MgCl_2_, 5 mM NaHCO_3_, 5 mM KCl, and 140 mM NaCl (pH 7.4)) and pelleted at 16,700× *g* for 10 min. For mass spectrometry samples, the pellet was washed by resuspension in ionic media a second time and pelleted at 10,000× *g* for 10 min. This final pellet was lysed directly by sonication in 100 mM Tris-HCl with 4% (*w*/*v*) SDS and 0.1 M DTT adjusted to pH 7.6 and heated to 95 °C for 5 min. For Seahorse experiments, the final pellet was resuspended in incubation media (IM; 3.5 mM KCl, 120 mM NaCl, 1.3 mM CaCl_2_, 0.4 mM KH_2_PO_4_, 1.2 mM Na_2_SO_4_, and 2 mM MgSO_4_ (pH 7.4 at 37 °C)). Protein concentration was determined by Pierce 660 nm Protein Assay. Lysates for mass spectrometry were stored at −80 °C until further processing. Seahorse samples were used fresh.

### 4.4. Seahorse Metabolic Flux

#### 4.4.1. Bioenergetic Analysis of Synaptosomes

Synaptic bioenergetic function was assessed using the Seahorse XFe96 analyzer following a workflow modified from the protocol established by Choi [84]. Briefly, after quantification by BCA, 15 µg of isolated synaptosomes were plated onto a poly-D-lysine-coated Seahorse cell culture plate (Agilent, Santa Clara, CA, USA) from a stock-normalized concentration of 1.5 mg/mL in IM. To ensure adherence of synaptosomes to the bottom, the plate was centrifuged at 2500× *g* for 30 min at room temp. Following centrifugation, IM containing 4 mg/mL BSA prewarned to 37 °C was added to bring well volume to 175 μL and 5 μL of IM containing 360 mM pyruvate, and 540 mM glucose was added for a final concentration of 10 and 15 mM, respectively. For the Seahorse experiments, synaptosomes were isolated from male mice aged 6 months (*n* = 13, 7, 8, and 5 for WT, Polg, PolgPKO, and PolgW402A, respectively). For each biological replicate, 5–6 technical replicate wells were used. Statistical analysis was performed in GraphPad Prism 9 (Version 9.0.0) using two-way ANOVA with Tukey’s correction for multiple comparisons. 

#### 4.4.2. Bioenergetic Analysis of Liver Mitochondria

Isolated hepatic mitochondria were assessed for mitochondrial function using the electron flow assay on a Seahorse XFe96 analyzer based on a protocol modified, as described previously [85], from that of Rogers [86]. For the electron flow experiments, liver mitochondria isolated (as described above) from male mice (*n* = 10, 4, 8, and 3 for WT, Polg, PolgPKO, and PolgW402A, respectively) at 6 months of age were used. Each biological replicate was comprised of 3–4 technical replicate wells for the experiment. Statistical analysis was performed in GraphPad Prism 9 (Version 9.0.0) using two-way ANOVA with Tukey’s correction for multiple comparisons. 

### 4.5. Mass Spectrometry

#### 4.5.1. Mass Spectrometry Sample Prep

Peptides for mass spectrometry were prepared by tryptic digest from mouse liver mitochondrial and synaptosomal protein lysates using the filter-aided sample preparation (FASP) method [87]. Peptides were desalted and enriched for positive charge using Oasis mixed-mode strong cation exchange cartridges (Waters) and subsequently dehydrated with a Savant ISS 110 SpeedVac concentrator. Dry peptides were solubilized with 0.1% formic acid (FA) prior to this and quantified by the Scopes [88] method with a NanoDrop 2000 UV–vis spectrophotometer (Wilmington, DE, USA) by absorbance at 205 nm. Mass spectrometry injection samples were prepared at 0.5 µg/µL in 2% acetonitrile (ACN) and 0.1% FA. 

#### 4.5.2. Mass Spectrometry Acquisition

To minimize batch effects in the collection of mass spectrometry data, the sample injection order was randomized in a manner that prioritized maintaining genotypes together while intermixing labeling periods such that for the liver mitochondria and synaptosomes, there was no sequence of 3 or 4 injections within each genotype. The randomization order is provided in Appendix A. Samples were blinded before submission to the UNMC Mass Spectrometry Core Facility for data acquisition and only de-blinded after running through data analysis software. 

##### Liquid Chromatography

For each sample, 2.5 µg was loaded onto a trap column with Acclaim PepMap 100 75 µm × 2 cm C18 LC Columns at a flow rate of 4 µL/min, followed by separation with a Thermo RSLC Ultimate 3000 on a Thermo Easy-Spray PepMap RSLC C18 75 µm × 50 cm C-18 2 μm column utilizing a step gradient of 9–25% solvent B (0.1% FA in 80% ACN) from 10–100 min and 25–45% solvent B for 100–130 min at 300 nL/min and 50 °C, with a 155 min total run time. 

##### Data Acquisition

Eluted peptides were analyzed by a Thermo Orbitrap Fusion Lumos Tribrid mass spectrometer in data-dependent acquisition (DDA) mode. Full-scan survey mass spectra were acquired in the Orbitrap at a resolution of 60,000 from 375 to 1500 *m*/*z*. The AGC target for MS1 was set at 100%, and an ion filling time at 50 ms. The most intense ions with charge states 2–5, isolated in 3 s cycles and fragmented using HCD fragmentation at 30% normalized collision energy, were detected at a mass resolution of 15,000. The AGC target for MS2 was set at 100%, and ion filling time at 22 ms, with a dynamic exclusion of 60 s and a 10 ppm mass window.

#### 4.5.3. Mass Spectrometry Analysis

##### Abundance

Protein expression (abundance) analysis was performed as we have previously reported [47]. Briefly, protein lysates were obtained from liver mitochondria (*n* = 8) or synaptosomes (*n* = 12) of 180-day-old (6-month-old) WT, Polg, PolgPKO, and PolgW402A male mice and quantified by Pierce 660 nm protein assay. Fifty (50) µg of protein was used to prepare peptides for mass spectroscopy by the FASP method, as previously described [87,89]. Peptides (2.5 µg) from each sample were injected for mass spectrometry analysis (see above). Protein identifications were obtained by searching raw data against the UniProt Mus musculus proteome (UP000000589) in MaxQuant (2.0.3.0) [90,91], with the internal contaminant list enabled. Two missed cleavages were allowed for trypsin/P enzymatic specificity, and label multiplicity was set to two, with a heavy label for tri-deuteroleucine defined (+3.0188325 Da). Methionine oxidation, phosphorylation (STY), and N-terminal acetylation were specified as variable modifications, with a single fixed modification of carbamidomethylation (C). Protein abundance was quantified using the MaxQuant [90,91] standard label-free quantification (LFQ) algorithm with default settings. Additionally, match between runs, second peptide, and dependent peptide search options were enabled with an FDR of 0.01. Heavy and light LFQ intensities were summed to yield the total LFQ intensity for each protein ID and sample combination before downstream data analysis. Total LFQ intensities were analyzed as previously [47] using LFQ-Analyst [92] as follows: total LFQ intensities were normalized, assuming most proteins are unchanged between groups. A custom-generated R script was used to conduct statistical analysis on the proteinGroups.txt file. First, the raw expression data were cleaned by filtering contaminants, reverse hits, those identified “only by site”, those identified by a single peptide, or those inconsistently identified or quantified in the same experimental condition. LFQ expression data were converted to log2 scale, grouped by condition, and the “Missing not At Random” method was used to impute missing values. This method relies on a random sampling of a left-shifted Gaussian distribution of 1.8 StDev apart, with a width of 0.3. Protein-wise linear models were combined with empirical Bayes statistics to determine differential expression. The initial density of valid values for search results is illustrated in Appendix A. Proteins were determined as differentially expressed by the R Bioconductor package, limma [93], using an adjusted p-value cutoff of 0.05 (Benjamini–Hochberg method) and an absolute log2 fold change of 1 in each pairwise comparison. Lists of proteins identified as exhibiting statistically significant differential expression by this method were further refined by employing a missing value cutoff requiring a minimum of 50% of samples from each genotype (i.e., ≥4 valid values for liver mitochondria or 6 valid values for synaptosome were required in every genotype). Differentially expressed proteins that were quantified in both tissues were hierarchically clustered by Euclidean distances with average linkage in Perseus (1.6.15.0) [94], with 10 row clusters and all other default settings. These clustered proteins were then filtered for those that were deemed significant in one or both tissues by quantitative analysis.

##### Turnover

Analysis of protein half-lives (turnover) was performed as we have previously reported [47]. Briefly, protein lysates were obtained from liver mitochondria (*n* = 3) or synaptosomes (*n* = 3) of 180-day-old (6-month-old) WT, Polg, PolgPKO, and PolgW402A male mice and quantified by Pierce 660 nm protein assay. Hepatic mitochondrial samples corresponding to metabolic labeling for 3, 8, and 15 days were subjected to mass spectrometry, and synaptosome samples corresponding to 8, 15, 24, and 35 days of metabolic labeling were analyzed. Fifty (50) µg of protein was used to prepares peptides for mass spectroscopy by the FASP method, as previously described [87,89]. Peptide (2.5 µg) from each sample was injected for mass spectrometry analysis (see above). Hardklor (v2.3.0) [95,96] and Bullseye (v1.30) [97] were used to conduct precursor feature refinement, followed by a database search by Comet (2018.01 rev. 2) [98] against a custom UniProt library composed of canonical proteins for Mus musculus (UP000000589) combined with common contaminant proteins. Cysteine carbamidomethylation was included as a +57.021461 Da static modification of cysteine, and methionine oxidation and tri-deuteroleucine were accounted for by the inclusion of dynamic modifications of +15.9949 Da and +3.0188325 Da, respectively. Enzymatic specificity was set to trypsin/P, and two missed cleavages were allowed, with a precursor mass tolerance of 10 ppm. Crux-Percolator (3.02.0) [99,100] was used to rescore FDR with reverse decoys. Proteins that passed an FDR q-value threshold of 0.01 were used to build a BlibBuild [101] spectrum sequence list format using a custom R script and imported into Topograph [48] for turnover calculations.

The Topograph-daily x64 release was used to calculate half-lives of proteins by incorporating data from all detected peptides. A minimum of 10 total values of percent newly synthesized protein per genotype and only unique peptides were included for analysis to limit ambiguity. The following quality controls in Topograph were used to control data quality: turnover score ≥ 0.98, deconvolution score ≥ 0.95, total area under the curve ≥ 1,000,000, and data points above or below the protein mean by more than 2 standard deviations. Data points for all biological replicates were pooled for half-life calculations, and those with excessive variability of percent newly synthesized were identified and omitted using a censure comparable to the coefficient of variation, as previously reported [47], where proteins with a 95% confidence interval/half-life ratio ≥0.3 were excluded from the analysis [13]. In liver mitochondria, this resulted in a total of 397, 517, 535, and 524 proteins for liver and 806, 1042, 1067, and 1048 proteins for synaptosomes with quantified half-lives in WT, Polg, PolgPKO, and PolgW402A, respectively. 

### 4.6. Pathway Analysis 

Log2 fold-change values for all proteins identified in either liver mitochondria or synaptosomes were interrogated by pathway analysis using Ingenuity Pathway Analysis (IPA; Qiagen, Ann Arbor, MI, USA) [101] to identify “Canonical Pathways” impacted by loss of Parkin or enhanced Parkin activity. Activation Z-scores were clustered hierarchically using average linkage Euclidean distance clustering in Perseus (1.6.15.0) [94], with 15 row clusters and all other default settings. 

### 4.7. Western Blot Analysis

Protein lysates prepared as described above were quantified using the Pierce 660 nm Protein Assay. Thirty (30) µg of total protein was resolved on 10% Tris gels using the Tris/Glycine buffer system and transferred to poly-vinylidene-fluoride membrane using the iBlot dry transfer system. After transfer, 3% non-fat milk in tris-buffered saline with 0.1% Tween-20 (TBST) was used to block membranes for 1 h at room temp. Primary antibody was diluted in 1% non-fat milk/TBST and incubated on membranes overnight (Parkin 5C3 1:200, BioLegend, San Diego, CA, USA (No.865602)). Before incubation with secondary antibody, membranes were washed 3 × 10 min with TBST. Secondary antibody (goat anti-mouse-HRP, 1:1000 Cell Signaling) was diluted in 0.5% non-fat milk-TBST and then incubated for 1 h at room temp, followed by 3 × 10 min washes with TBST. Blot images were acquired on a Licor Odyssey at 10 min per channel and quantified by densitometry. Target protein expression was normalized to total protein loading via Coomassie staining, and statistical analysis was performed in GraphPad Prism 9 by one-way ANOVA.

### 4.8. Statistical Analysis

Statistical analysis of turnover data was performed in processing using Topograph according to the conditions outlined in Section 4.5.3 to calculate half-lives. Statistical analysis of bulk abundance data was achieved as described in Section 4.5.3; however, for individual measured proteins one-way ANOVA with Tukey’s post-test was used to determine the significance using Welch’s correction when sample sizes were not equivalent. For bioenergetic analysis, two-way ANOVA was used with Tukey’s post-test to correct for multiple comparisons. All data were visually inspected; however, data were retained inclusively regardless of apparent outlier status. 

## Figures and Tables

**Figure 1 ijms-25-06441-f001:**
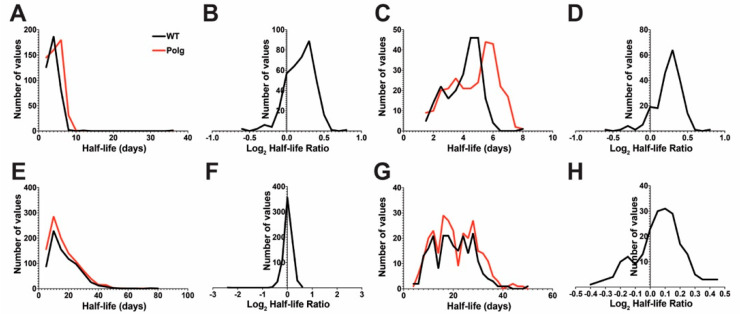
Impaired mtDNA quality control impacts proteome dynamics in liver mitochondria. Topograph was used to determine the half-lives of (**A**–**D**) liver mitochondria or (**E**–**H**) synaptosomes isolated from WT or Polg male mice at 6 months of age. Distributions of complete protein list half-lives (**A**,**E**). The MitoCarta 3.0 database was used to subset the protein list to those annotated as mitochondrial (**C**,**D**,**G**,**H**). Distributions of mitochondrial proteins list half-lives (**C**,**G**). Distribution of corresponding log2 fold change ratios (**B**,**D**,**F**,**H**).

**Figure 2 ijms-25-06441-f002:**
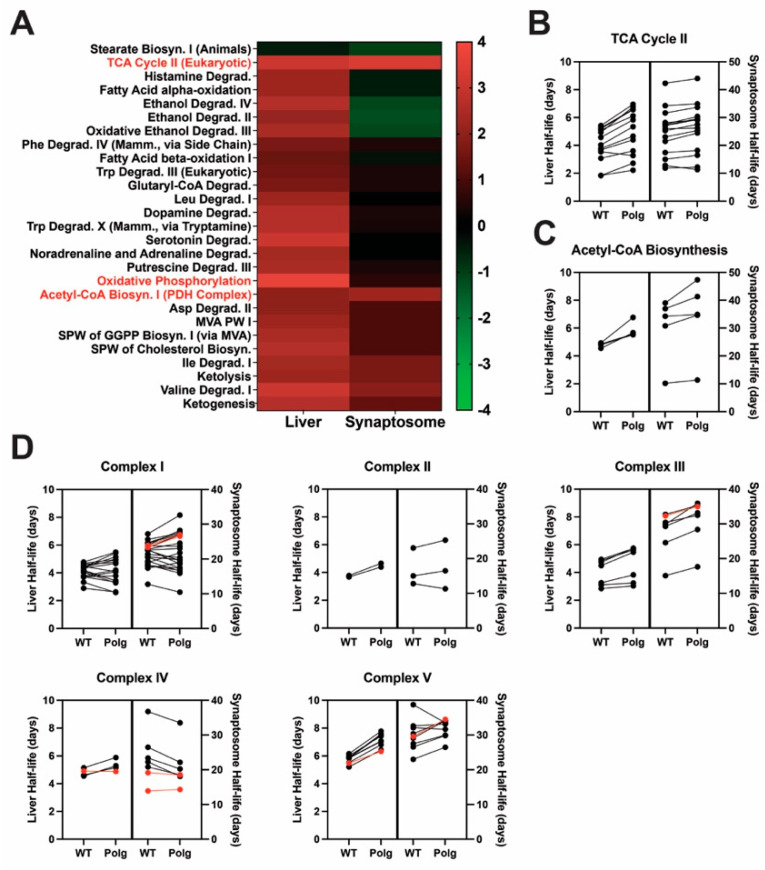
Polg mutation results in slower turnover of pathways of energy production. (**A**) Ingenuity Pathway Analysis (IPA) was used to assess Polg-driven changes in protein half-lives on a pathway scale for “Canonical Pathways” of “Metabolism”. Predicted activation Z-scores are presented as a heatmap hierarchically clustered by row using Euclidean distances and average linkage. (**B**–**D**) Half-lives (days) of all individual proteins analyzed in “Canonical Pathways” of “Metabolism” for (**B**) “TCA Cycle II (Eukaryotic)”, (**C**) “Acetyl-CoA Biosynthesis (Pyruvate Dehydrogenase Complex)”, and (**D**) “Oxidative Phosphorylation” (further subdivided by individual respiratory complex). Red data points in (**D**) represent mtDNA-encoded subunits detected/quantified. Polg-dependent increased half-lives, red.

**Figure 3 ijms-25-06441-f003:**
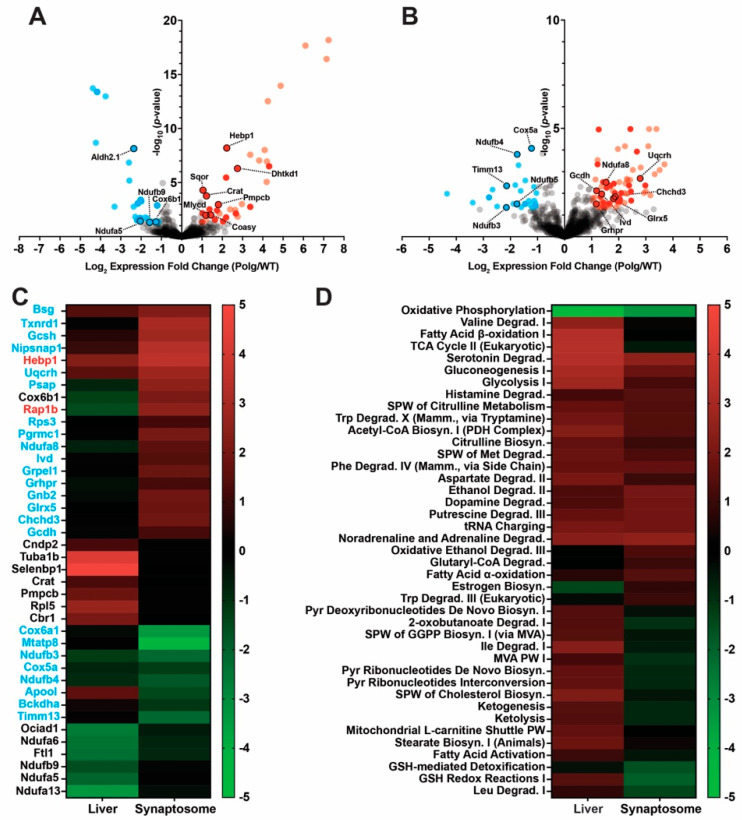
Loss of mitochondrial DNA fidelity alters the expression of electron transport chain components. Label-free quantitative proteomics analysis was used to identify differentially expressed (DE) proteins in (**A**) liver mitochondria and (**B**) synaptosomes isolated from 6-month-old male WT or Polg mice. Grey-non-DE proteins, blue-down-regulated, red-up-regulated (light-fill-fail 50% missing value threshold), black outline-proteins annotated as mitochondrial by MitoCarta 3.0. (**C**) DE proteins (in either tissue) were clustered hierarchically using Euclidean distances with average linkage. Text color represents significance. Red-both tissues, blue-synaptosomes, black-liver mitochondria. IPA was used to predict changes in “Canonical Pathways” of (**D**) “Metabolism”. Activation Z-scores were hierarchically clustered using Euclidean distances with average linkage.

**Figure 4 ijms-25-06441-f004:**
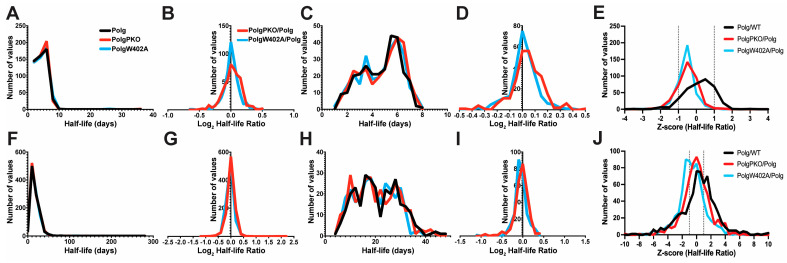
Modulation of Parkin impacts turnover of a subset of proteins. Topograph was used to determine the half-lives of (**A**–**E**) liver mitochondria or (**F**–**J**) synaptosomes isolated from 6-month-old Polg, PolgPKO, or PolgW402A male mice. Half-life distributions of complete protein lists (**A**,**F**). Half-life distributions of protein lists annotated as mitochondrial by the MitoCarta 3.0 database (**C**,**D**,**H**,**I**). Distributions of mitochondrial proteins list half-lives (**C**,**H**). Distribution of corresponding log2 fold change ratios (**B**,**D**,**G**,**I**). Distributions of Z-scores of half-lives between genotypes for each shared protein (**E**,**J**).

**Figure 5 ijms-25-06441-f005:**
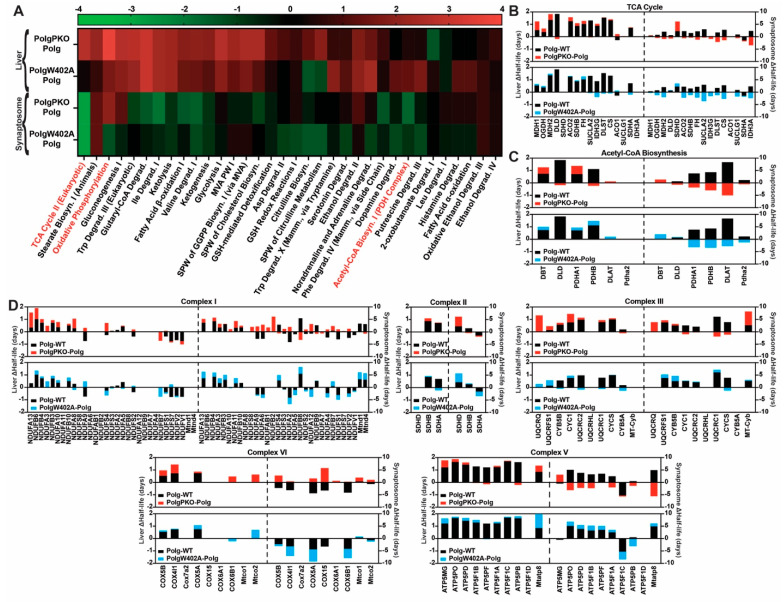
Pathway analysis reveals loss of mtDNA fidelity as the primary driver of global changes in protein turnover rates. IPA was used to predict large scale changes in protein turnover rates for Canonical Pathways of (**A**) “Metabolism”. Heatmaps represent Euclidean clustered pathway activation Z-scores for the double mutant/Polg comparison, with average linkage for the indicated double mutant. (**C**,**D**) Comparison of the difference in half-life (ΔHalf-life) for the indicated pairwise comparison of all proteins with quantified half-lives for the “Canonical Path-ways” of “Metabolism” (**B**) “TCA Cycle II (Eukaryotic)”, (**C**) “Acetyl-CoA Biosynthesis (Pyruvate Dehydrogenase Complex)”, and (**D**) “Oxidative Phosphorylation” (subdivided by respiratory complex). Bars represent the difference between the half-life of the indicated double mutant and WT.

**Figure 6 ijms-25-06441-f006:**
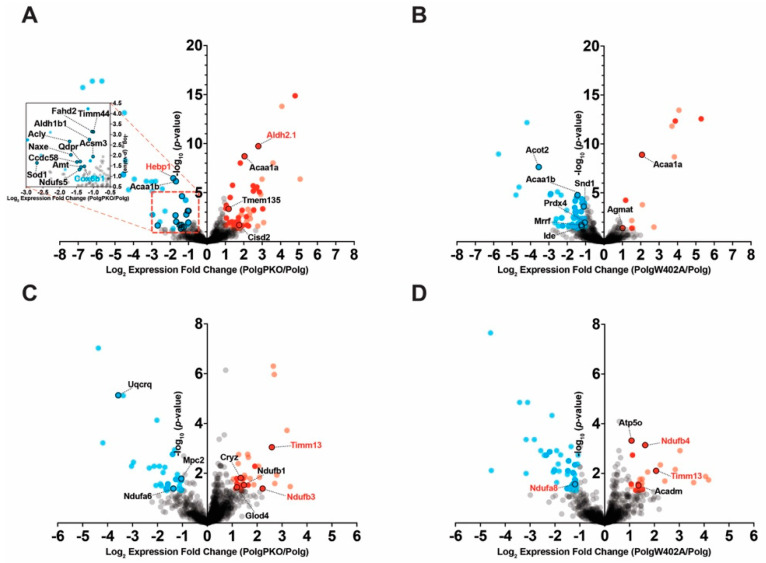
The liver mitochondrial proteome is more susceptible to loss- or gain-of-function Parkin mutations under elevated mtDNA stress. Label-free quantitative proteomics analysis was used to identify differentially expressed (DE) proteins identified in the indicated double mutant to single mutant (Polg only) comparison from (**A**,**B**) liver mitochondria or (**C**,**D**) synaptosomes isolated from 6-month-old male WT or Polg mice. Gray dots-non-DE proteins, blue-down-regulated, red-up-regulated (light-filled dots-fail 50% missing value threshold), black circles-proteins annotated as mitochondrial by Mito-Carta 3.0. Mitochondrially annotated protein names in red represent those exhibiting expression directionality reversal compared to the respective Polg/WT comparison (see Figure 3), and names in blue text represent proteins exhibiting expression synergism.

**Figure 7 ijms-25-06441-f007:**
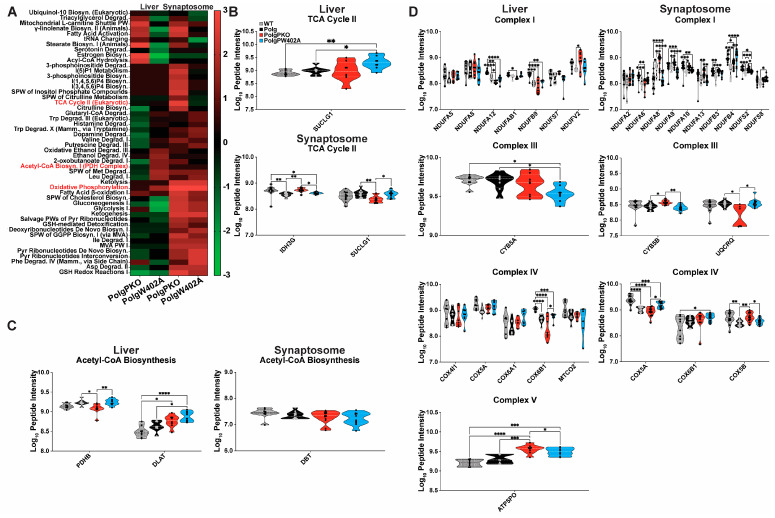
Loss- or gain-of-function Parkin mutations influence expression of proteins that contribute to bioenergetics. Log2 fold change expression ratios were analyzed by IPA to predict functional outcomes of loss- or gain-of-function Parkin mutation amongst the “Canonical Pathways” of (**A**) “Metabolism”. Heatmaps represent only Euclidean clustered activation Z-scores for the indicated double mutant/Polg comparison with average linkage. (**B**–**D**) Log10-transformed raw peptide intensities for the indicated proteins and strain for proteins exhibiting a minimum of |0.5| log2 fold change for double mutant to Polg comparison and passed a missing value threshold of at least 50% valid values for each genotype (i.e., four valid values in each genotype for liver mitochondria and six valid values for each genotype for synaptosomes) in the Canonical Pathways (**B**) “TCA Cycle II (Eukaryotic)”, (**C**) “Acetyl-CoA Biosynthesis (Pyruvate Dehydrogenase Complex)”, and (**D**) “Oxidative Phosphorylation” (by respiratory complex). Truncated violin plots contain all individual points of data, and statistical significance was determined using the raw peptide intensities’ pre-log transformation on a per protein basis by ordinary one-way ANOVA with Tukey’s multiple comparison correction. * *p* < 0.05, ** *p* < 0.01, *** *p* < 0.001, **** *p* < 0.0001.

**Figure 8 ijms-25-06441-f008:**
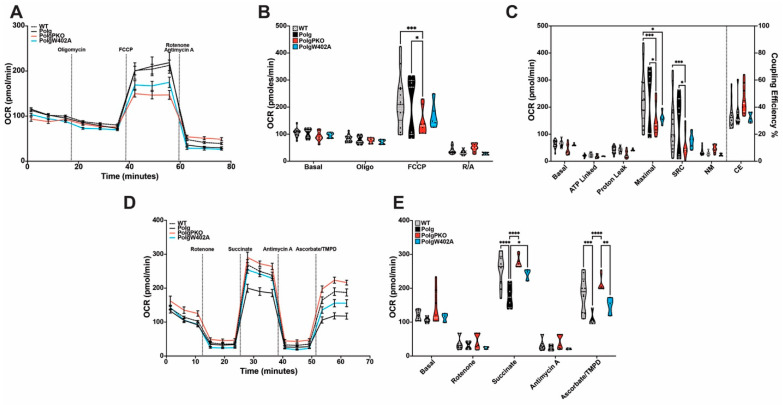
Parkin status and loss of mtDNA fidelity differentially affect bioenergetics of synaptosomes and liver mitochondria. Synaptosomes (**A**–**C**) or liver mitochondria (**D**,**E**) were isolated from 6-month-old male WT, Polg, PolgPKO, or PolgW402A mice. The Mito Stress Test (**A**,**B**) was used to assess synaptic bioenergetics. (**C**) Respiration states as calculated from oxygen consumption rate for synaptosomes. The Electron Flow Assay (**D**,**E**) was employed to determine liver mitochondrial respiration. Truncated violin plots contain points representative of the individual biological replicate values. For the Mito Stress Test, n = 13 (WT), 7 (Polg), 8 (PolgPKO), 5 (PolgW402A), and each biological replicate was comprised of 4–6 technical replicate wells. For the Electron Flow Assay, n = 10 (WT), 4 (Polg), 8 (PolgPKO), 3 (PolgW402A), and each biological replicate was comprised of 2–3 technical replicate wells. Statistical significance was determined using two-way ANOVA with Tukey’s multiple comparison test to compare the mean of every column within each row. * *p* < 0.05, ** *p* < 0.01, *** *p* < 0.001, **** *p* < 0.0001 (NM—non-mitochondrial; SRC—spare respiratory capacity; CE—coupling efficiency).

## Data Availability

Data are contained within the article and Appendix A.

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
