# Peer review of "Mitochondrial DNA Instability Supersedes Parkin Mutations in Driving Mitochondrial Proteomic Alterations and Functional Deficits in Polg Mutator Mice"

_ijms, 2024, doi:10.3390/ijms25126441_

Round 1

Reviewer 1 Report

Comments and Suggestions for Authors

The present paper titled "Mitochondrial DNA instability supersedes Parkin mutations in driving mitochondrial proteomic alterations and functional  deficits in Polg mutator mice" is a very well written and interesting paper with he potential to advance our understanding of Parkin function and mitochondrial quality control in the context of two tissues: mitochondria and synaptosomal preparations in neurons by using mouse models of mitochondrial quality control and DNA repair. The findings are very well described, rigorously designed and meticulously analyzed. The background and introduction is succinct and provides a good premise for the study and cites sufficient number of relevant papers. The data is robust including the IPA analysis of protein turn over as well as the bioenergetic readouts.

While the manuscript is well written and the authors give many valuable insights, the  copious amount of proteomic, pathway and biochemical data presented, and data presentation makes it challenging for reviewers and readers to appreciate the key findings and parse the data in an efficient manner. Some of the results are contradictory among the different genotypes and crossings of mice which makes the interpretation and discussion of the data challenging. To strengthen this interesting paper, some modest and a few major concerns are listed below to help the authors improve the quality of the paper:

1) While it is clear that the POLG mutator mice shows a detrimental reduction in weight and increased turnover (increased half lives) and increase expression or levels of mitochondrial liver proteins vs. synaptic proteins (figures 1-3), the POLG/Parkin knockout mice shows a very mild to non-significant phenotype in affecting protein turnover in Figure 4 and is not convincing. The authors state that the PolgPKO/PolgPKO shows an increase in the half-life of proteins in liver mitochondria but the line graphs show a similar overlap to PolG alone in panel A. In panel B, the PolgPKO vs. Polg W402A shows a modest reversal in half life or left shift. It is very hard to appreciate this differences especially if a statistical analysis on the extent of this shift is not provided. A Z score or statistical analysis not provided to know if the effect of loss of Parkin or over-activation (disinhibition) is significant. A more detailed analysis or tempering down the interpretation of the data is needed.

2) Along the same line, since Parkin is expressed in all tissues including brain tissue, what is the level of expression of Parkin in synaptosomes compared to liver based on proteomic analysis (peptide counts)? Is the negative effect of the loss of Parkin on protein turnover in synaptosomes compared to liver due to a low expression of Parkin relative to liver? 

3) For figure 5, there is a large amount of data in the bar graphs which makes it parse and of the phenotypes are contradictory in terms of the genotypes of the mice even within the same class of proteins being analyzed (TCA cycle or complex II/III). For instance, while it is clear that a loss of Parkin function can lead to a slowed turnover and increased expression of mitochondrial proteins as shown for MDH1 and MDH2 in liver mitochondria in the TCA category (Figure 4, panel B) which is consistent with a ubiquitin/proteasome function of Parkin. However, it is not clear while many other proteins are downregulated in the synaptosomes. Over-activating Parkin in the double mutant Pol/PKO-PolG mice leads to a decrease or reversal of some proteins as expected (IDH3G in liver) and as pointed by the authors but many other proteins are not (decreased) or stay elevated which is not consistent with overactivation of Parkin. Is it possible that certain proteins that are reverse Parkin substrates whereas other ones that are elevated are due to a non-specific effect or pleotropic effect of loss of Parkin? The best type of experiment will be to complement Parkin back (breeding) in the PolGPKO-PolG to wild-type and see which proteins are reversed. Was this scenario considered? I know there are many data points not consistent or contrary to expected results based on genotype but the authors should provide a stronger rationale for these disparate differences in some data sets.

4) The authors mention that glycolysis is expected to be overactivated due to loss of Parkin in the discussion. However, the authors do not point to the data in figure 5 panel A which shows that glycolysis proteins like glycolysis I and others are increased in the Parkin null/ PolG mutator-KO mice nor discussed in more detail. This would be useful to discuss in the context of the proteomic data presented if glycolysis is upregulated to compensate for mitochondrial dysfunction due to loss of Parkin and POLG.

5) While the data in figure 6 is straightforward to interpret, the same cannot be said for figure 7. The loss of POLG shows consistent increases in many mitochondrial proteins. However,  it is not clear why will loss of Parkin leads to a significant decrease in complex I subunits whereas others are unchanged or remain the same. Again, are these proteins that are increased by loss of Parkin and reversed with overactivation (disinhibition) specific substrates (localized degradation by ubiquitination) of Parkin whereas others are altered due to non-specific or indirect effects of loss of Parkin?

Localization conundrum (Parkin): Even if these proteins are affected by Parkin activity, how is it possible that complex proteins in the inner mitochondrial membrane is affected by Parkin if Parkin is mostly cytosolic and binds the outer mitochondrial membrane? 

6) For figure 8, the authors provided a good rationale for the contradictory phenotypic differences due to Parkin loss in terms of the bioenergetics data. However, a loss of Parkin function is known to cause loss of mitochondrial biogenesis, and mitochondrial dysfunction which is not consistent with a bioenergetic rescue observed in figure 8. Parkin is involved in mitochondrial quality control and "rejuvenating mitochondria via mitophagy/biogenesis). Is it possible that the rescue of mitochondrial dysfunction  in figure 8 D-E due to loss of Parkin in POLG mice is not related to Parkin?

7) Finally, given that Parkin regulates mitophagy, did the authors observed any alterations in the expression levels of autophagy proteins (LC3 in mitochondria)  or the autophagic substrate P63 in the liver and brain?  One can expect that POLG mice to have overactive mitophagy to remove dysfunctional mitochondrial and mutated DNA.  This may help reconcile whether some of the effects of Parkin is due to its ubiquitin-ligase activity vs. induction of mitophagy.

Minor point: In figure 6, there is a possible mistake in referring to data. The authors state that A-B shows plots of the mitochondrial proteome whereas B-D is synaptosomes. Did the authors meant that C-D point to synaptosomes? There are other potential inconsistencies in the figure legends in other figures so I will encourage the authors to check all figure references carefully.

Comments on the Quality of English Language

It is fine.

Reviewer 2 Report

Comments and Suggestions for Authors

The manuscript is well written and describes findings from examining the Mutator and Mutator mouse crossed with a Parkin KO and Parkin mutant mouse that cannot be dis-inhibited.  The group performs MS analysis to determine the turnover rate of proteins for these animals at 6 months of age comparing liver and synaptosomes.   Respiration was also examined. The only a few concerns for the manuscript. One was figure presentation (Figure 7 is too small to read). The other concern is that the discussion should be put in the framework or context of the age of animals analyzed here from what is known from the literature about Polg mice, mitochondrial dysfunction, and development of phenotypes for liver and brain at different ages analyzed in peer reviewed published work.    

Comments on the Quality of English Language

No comment on quality of English. It is fine. 

Round 2

Reviewer 1 Report

Comments and Suggestions for Authors

The authors did a good job in addressing most of my concerns to the extent of their abilities including providing additional statistical data like Z scores and providing additional supplemental data which provided additional rigor to the proteomic data and its interpretation. All other points were addressed in the discussion regarding alternative pathways that exists that are Parkin-independent.

However, I noticed a few typos in the revised manuscript which the authors should consider revising again including "promximity".

Also, my request for correcting the labeling in figure 6 (should be C-D instead of referring to panels B-D) appears to not be corrected.

If this can be corrected before publication then everything should be OK. Other than that, the paper is very thorough and comprehensive in the analysis, and significance of the data presented.

Comments on the Quality of English Language

Needs some careful revision prior to publication. For instance,  I noticed a few typos in the revised manuscript which the authors should consider revising again including "promximity".
